# On the Relationship Between LFP & Spiking Data

**David E. Carlson[1], Jana Schaich Borg[2], Kafui Dzirasa[2], and Lawrence Carin[1]**
[1]Department of Electrical and Computer Engineering
[2]Department of Psychiatry and Behavioral Sciences
Duke University
Duham, NC 27701
{david.carlson, jana.borg, kafui.dzirasa, lcarin}@duke.edu

## Abstract

One of the goals of neuroscience is to identify neural networks that correlate with important behaviors, environments, or genotypes. This work proposes a strategy for identifying neural networks characterized by time- and frequency-dependent connectivity patterns, using convolutional dictionary learning that links spike-train data to local field potentials (LFPs) across multiple areas of the brain. Analytical contributions are: ($i$) modeling dynamic relationships between LFPs and spikes; ($ii$) describing the relationships between spikes and LFPs, by analyzing the ability to predict LFP data from one region based on spiking information from across the brain; and ($iii$) development of a clustering methodology that allows inference of similarities in neurons from multiple regions. Results are based on data sets in which spike and LFP data are recorded simultaneously from up to 16 brain regions in a mouse.

## 1 Introduction

One of the most fundamental challenges in neuroscience is the "large-scale integration problem": how does distributed neural activity lead to precise, unified cognitive moments [1]. This paper seeks to examine this challenge from the perspective of extracellular electrodes inserted into the brain. An extracellular electrode inserted into the brain picks up two types of signals: (1) the local field potential (LFP), which represents local oscillations in frequencies below 200 Hz; and (2) single neuron action potentials (also known as "spikes"), which typically occur in frequencies of 0.5 kHz. LFPs represent network activity summed over long distances, whereas action potentials represent the precise activity of cells near the tip of an electrode. Although action potentials are often treated as the "currency" of information transfer in the brain, relationships between behaviors and LFP activity can be equally precise, and sometimes even more precise, than those with the activity of individual neurons [2, 3]. Further, LFP network disruptions are highly implicated in many forms of psychiatric disease [4]. This has led to much interest in understanding the mechanisms of how LFPs and action potentials interact to create specific types of behaviors. New multisite recording techniques that allow simultaneous recordings from a large number of brain regions provide unprecedented opportunities to study these interactions. However, this type of multi-dimensional data poses significant challenges that require new analysis techniques.

Three of the most challenging characteristics of multisite recordings are that: 1) the networks they represent are dynamic in space and time, 2) subpopulations of neurons within a local area can have different functions and may therefore relate to LFP oscillations in specific ways, and 3) different frequencies of LFP oscillations often relate to single neurons in specific ways [5]. Here new models are proposed to examine the relationship between neurons and neural networks that accommodate these characteristics. First, each LFP in a brain region is modeled as convolutions between a bounded-time dictionary element and the observed spike trains. Critically, the convolutional factors are allowed to be dynamic, by binning the LFP and spike time series, and modeling the dictionary element for

each bin of the time series. Next, a clustering model is proposed making each neuron's dictionary element a scaled version of an autoregressive template shared among all neurons in a cluster. This allows one to identify sub-populations of neurons that have similar dynamics over their functional connectivity to a brain region. Finally, we provide a strategy for exploring which frequency bands characterize spike-to-LFP functional connectivity. We show, using two novel multi-region electrophysiology datasets from mice, how these models can be used to identify coordinated interactions within and between different neuronal subsystems, defined jointly by the activity of single cells and LFPs. These methods may lead to better understanding of the relationship between brain activity and behavior, as well as the pathology underlying brain diseases.

## 2 Model

### 2.1 Data and notation

The data used here consists of multiple LFP and spike-train time series, measured simultaneously from $M$ regions of a mouse brain. Spike sorting is performed on the spiking data by a VB implementation of [6], from which $J$ single units are assumed detected from across the multiple regions (henceforth we refer to single units as "neurons"); the number of observed neurons $J$ depends on the data considered, and is inferred as discussed in [6]. Since multiple microwires are inserted into single brain regions in our experiments (described in [7]), we typically detect between 4-50 neurons for each of the $M$ regions in which the microwires are inserted (discussed further when presenting results). The analysis objective is to examine the degree to which one may relate (predict) the LFP data from one brain region using the $J$-neuron spiking data from all brain regions. This analysis allows the identification of multi-site neural networks through the examination of the degree to which neurons in one region are predictive of LFPs in another.

Let $\boldsymbol{x} \in \mathbb{R}^T$ represent a time series of LFP data measured from a particular brain region. The $T$ samples are recorded on a regular grid, with temporal interval $\Delta$. The spike trains from $J$ different neurons (after sorting) are represented by the set of vectors $\{\boldsymbol{y}_1, \ldots, \boldsymbol{y}_J\}$, binned in the same manner temporally as the LFP data. Each $\boldsymbol{y}_j \in \mathbb{Z}_+^T$ is reflective of the number of times neuron $j \in \{1, \ldots, J\}$ fired within each of the $T$ time bins, where $\mathbb{Z}_+$ represents nonnegative integers.

In the proposed model LFP data $\boldsymbol{x}$ are represented as a superposition of signals associated with each neuron $\boldsymbol{y}_j$, plus a residual that captures LFP signal unrelated to the spiking data. The contribution to $\boldsymbol{x}$ from information in $\boldsymbol{y}_j$ is assumed generated by the convolution of $\boldsymbol{y}_j$ with a bounded-time dictionary element $\boldsymbol{d}_j$ (residing within the interval $-L$ to $L$, with $L \ll T$). This model is related to convolutional dictionary learning [8], where the observed (after spike sorting) signal $\boldsymbol{y}_j$ represents the signal we convolve the learned dictionary $\boldsymbol{d}_j$ against.

We model $\boldsymbol{d}_j$ as time evolving, motivated by the expectation that neuron $j$ may contribute differently to specified LFP data, based upon the latent state of the brain (which will be related to observed animal activity). The time series $\boldsymbol{x}$ is binned into a set of $B$ equal-size contiguous windows, where $\boldsymbol{x} = \text{vec}([\boldsymbol{x}_1, \ldots, \boldsymbol{x}_B])$, and likewise $\boldsymbol{y} = \text{vec}([\boldsymbol{y}_{j1}, \ldots, \boldsymbol{y}_{jB}])$. The dictionary element for neuron $j$ is similarly binned as $\{\boldsymbol{d}_{j1}, \ldots, \boldsymbol{d}_{jB}\}$, and the contribution of neuron $j$ to $\boldsymbol{x}_b$ is represented as a convolution of $\boldsymbol{d}_{jb}$ and $\boldsymbol{y}_{jb}$. This bin size is a trade-off between how finely time is discretized and the computational costs.

In the experiments, in one example the bins are chosen to be 30 seconds wide (novel-environment data) and in the other 1 minute (sleep-cycle data), and these are principally chosen for computational convenience (the second data set is nine times longer). Similar results were found with windows as narrow as 10 second, or as wide as 2 minutes.

### 2.2 Modeling the LFP contribution of multiple neurons jointly

Given $\{\boldsymbol{y}_1, \ldots, \boldsymbol{y}_J\}$, the LFP voltage at time window $b$ is represented as

$$\boldsymbol{x}_b = \sum_{j=1}^{J} \boldsymbol{y}_{jb} * \boldsymbol{d}_{jb} + \boldsymbol{\epsilon}_b \tag{1}$$

where $*$ represents the convolution operator. Let $\mathbf{D}_j = [\boldsymbol{d}_{j1}, \ldots, \boldsymbol{d}_{jB}] \in \mathbb{R}^{(2L+1) \times B}$ represent the sequence of dictionary elements used to represent the LFP data over the $B$ windows, from the perspective of neuron $j$. We impose the clustering prior

$$\mathbf{D}_j = \zeta_j \mathbf{A}_j, \quad \mathbf{A}_j \sim G, \quad G \sim \text{DP}(\beta, G_0) \tag{2}$$

where $G$ is a draw from a Dirichlet process (DP) [9, 10], with scale parameter $\beta > 0$ and base probability measure $G_0$. Note that we cluster the *shape* of the dictionary elements, and each neuron has its own scaling $\zeta \in \mathbb{R}$. Concerning the base measure, we impose an autoregressive prior on the temporal dynamics, and therefore $G_0$ is defined by an AR($\alpha, \gamma$) process

$$\boldsymbol{a}_b = \alpha \boldsymbol{a}_{b-1} + \boldsymbol{\nu}_t, \;\; \boldsymbol{\nu}_t \sim \mathcal{N}(0, \gamma^{-1}\mathbf{I}) \tag{3}$$

where $\mathbf{I}$ is the identity matrix. This AR prior is used to constitute the $B$ columns of the DP "atoms" $\mathbf{A}_h^* = (\boldsymbol{a}_{h1}^*, \dots, \boldsymbol{a}_{hB}^*)$, with $G = \sum_{k=1}^{\infty} \pi_k \delta_{\mathbf{A}_k^*}$. The elements of the vector $\boldsymbol{\pi} = (\pi_1, \pi_2, \dots)$ are drawn from the "stick-breaking" [9] process $\pi_h = V_h \prod_{i<h}(1 - V_i)$ with $V_h \sim \mathrm{Beta}(1, \beta)$. We place the prior $\mathrm{Gamma}(a_\beta, b_\beta)$ on $\beta$, and priors Uniform(0,1) and $\mathrm{Gamma}(a_\gamma, b_\gamma)$ respectively on $\alpha$ and $\gamma$. To complete the model, we place the prior $\mathcal{N}(0, \tau^{-1}\mathbf{I})$ on $\boldsymbol{\epsilon}_b$, and $\zeta_j \sim \mathcal{N}(0, 1)$.

In the implementation, a truncated stick-breaking representation is employed for $G$, using $K$ "sticks" ($V_K = 1$), which simplifies the implementation and has been shown to be effective in practice [9] if $K$ is made large enough, and the size of $K$ is inferred during the inference algorithm.

Special cases of this model are clear. For example, if the $\mathbf{A}_j$ are simply drawn i.i.d. from $G_0$, rather than from the DP, each neuron is allowed to contribute its own unique dictionary shape to represent $\boldsymbol{x}_b$, called a *non-clustering* model in the results. In [11] the authors considered a similar model, but the time evolution of $\boldsymbol{d}_j$ was not considered (each neuron was assumed to contribute in the same way to represent the LFP, independent of time). Further, in [11] only a single neuron was considered, and therefore no clustering was considered. A multi-neuron version of this model is inferred by setting $B = 1$.

## 3 Inference

### 3.1 Mean-field Variational Inference

Letting $\Theta = \{\boldsymbol{z}, \boldsymbol{\zeta}, \mathbf{A}_{1,\dots,K}, V_{1,\dots,K}, \boldsymbol{\beta}, \alpha, \gamma\}$, the full likelihood of the clustering model

$$p(\boldsymbol{x}, \Theta) = \prod_{b=1}^{B} [p(\boldsymbol{x}_b|\Theta)] \prod_{j=1}^{J} [p(z_j|\boldsymbol{\pi})p(\zeta_j)] \prod_{k=1}^{K} [p(\mathbf{A}_k^*|\alpha, \gamma)p(V_k|\beta)] \, p(\beta, \alpha, \gamma) \tag{4}$$

The non-clustering model can be recovered by setting $z_j = \delta_j$ and the truncation level in the stick-breaking process $K$ to $J$. The time-invariant model is recovered by setting the number of bins $B$ to 1, with or without clustering. The model of [11] is recovered by using a single bin and a single neuron.

Many recent methods [12, 13] have been proposed to provide quick approximations to the Dirichlet process mixture model. Critically, in these models the latent assignment variables are conditionally independent when the DP parameters are given. However, in the proposed model this assumption does not hold because the observation $\boldsymbol{x}$ is the superposition of the convolved draws from the Dirichlet process.

A factorized variational distribution $q$ is proposed to approximate the posterior distribution, and the non-clustering model arises as a special case of the clustering model. The inference to fit the distribution $q$ is based on Bayesian Hierarchical Clustering [13] and the VB Dirichlet Process Split-Merge method [12]. The proposed model does not fit in either of these frameworks, so a method to learn $K$ by merging clusters by adapting [12, 13] is presented in Section 3.1.1. The factorized distribution $q$ takes the form:

$$q(\Theta) = \prod_j \left[ q(z_j) \prod_k q(\boldsymbol{\zeta}_{jk}) \right] q(\beta)q(\alpha)q(\gamma) \prod_k [q(\mathbf{A}_k^*)q(V_k)] \tag{5}$$

Standard forms on these distributions are assumed, with $q(z_j) = \mathrm{Categorical}(\boldsymbol{r}_j)$, $q(\gamma) = \mathrm{Gamma}(a_\gamma', b_\gamma')$, $q(\alpha) = \mathcal{N}_{(0,1)}(\hat{\alpha}, \eta_\alpha^{-1})$, $q(\mathbf{A}_k) = \mathcal{N}(\mathrm{vec}(\mathbf{A}_k); \mathrm{vec}(\hat{\boldsymbol{a}}_{k1}, \dots, \hat{\boldsymbol{a}}_{kB}), \boldsymbol{\Lambda}_k^{-1})$, $\boldsymbol{\Sigma}_k = \boldsymbol{\Lambda}_k$, and $q(\beta) = \mathrm{Gamma}(a_\beta', b_\beta')$. To facilitate inference, the distribution on $\zeta_j$ is split into $q(\zeta_{jk}) = \mathcal{N}(\mu_{jk}, \eta_{jk}^{-1})$, the variational distribution for $\zeta$ on the $j^{\text{th}}$ spike train given that it is in cluster $k$. The non-clustering model can be represented as a special case of the clustering model where $q(\zeta_{jk}) = \delta_1$, and $q(z_j) = \delta_j$. As noted in [12], this factorized posterior has the property that a $q$ with $K'$ clusters is nested in a representation of $q$ for $K$ clusters for $K \geq K'$, so any number of clusters up to $K'$ is represented.

Variational algorithms find a $q$ that minimizes the KL divergence from the true, intractable posterior [14], finding a $q$ that locally maximizes the evidence lower bound (ELBO) objective:

$$\log p(\boldsymbol{x}|\Theta) \geq \mathcal{L}(q) = \mathbb{E}_q[\log p(\boldsymbol{x}, \boldsymbol{z}, \boldsymbol{\zeta}, \mathbf{A}^*_{1,\dots,K}, \boldsymbol{\beta}, \alpha, \gamma|\Theta) - \log q(\boldsymbol{z}, \boldsymbol{\zeta}, \mathbf{A}^*_{1,\dots,K}, \boldsymbol{\beta}, \alpha, \gamma)] \quad (6)$$

To facilitate inference, approximations to $p(y|\Theta)$ are developed. Let $T_b$ be the number of time points in bin $b$, and define $\mathbf{R}_{jib} \in \mathbb{R}^{(2L+1)\times(2L+1)}$ with entries $R_{jib,ik} = \frac{1}{T_b}\sum_{t=1}^{T_b} y_{jb,t} y_{ib,t+k-i}$; $y_{jb,t}$ is $\boldsymbol{y}_j$ at time point $t$ in window/bin $b$. Let $\boldsymbol{x}_b^{-j} = \boldsymbol{x}_b - \sum_{j'\neq j} \boldsymbol{y}_b * (\sum_k r_{jk}\mu_{jk}\hat{\boldsymbol{a}}_{kb})$, or the residual after all but the contribution from the $j^{\text{th}}$ neuron have been removed, and define let $\boldsymbol{\nu}_{jb}^{-j} \in \mathbb{R}^{2L+1}$ with entries $\nu_{ji}^j = \frac{1}{T_b}\sum_{t=1}^{T_b} y_{jb,t} x_{b,t+i}$ for $i \in \{-L,\dots,L\}$. Both $\mathbf{R}_{jb}$ and $\nu_{jb}$ can be efficiently estimated with the FFT. For each time bin $b$, we can write: $\log p(\boldsymbol{x}_b^{-j}|\boldsymbol{y}_{jb}, \boldsymbol{d}_{jb}) = \text{const} - \frac{\tau}{2}(x_{b,t}^{-j} - \sum_{\ell=-L}^{L} y_{j,b,t+\ell} d_{j,b,-\ell})^2 \simeq \text{const} - \frac{\tau T_b}{2}(\boldsymbol{d}_{jb}^T \mathbf{R}_{jjb}\boldsymbol{d} - 2(\boldsymbol{\nu}_{jb}^{-j})^T \boldsymbol{d}_{jb})$

To define the key updates, let $\boldsymbol{y}'_{kb} = \sum_j r_{jk}\mu_{jk}\boldsymbol{y}_{jb}$, and $\boldsymbol{x}_b^{-k} = \boldsymbol{x}_b - \sum_{j'\neq j}\boldsymbol{y}'_{kb} * \hat{\boldsymbol{a}}_{kb}$. $\boldsymbol{\Sigma}_{kbb'}$ denotes the block in $\boldsymbol{\Sigma}_k$ indexing the $b$ and the $b'$ bins, which is efficiently calculated because $\boldsymbol{\Sigma}_k^{-1}$ is a block tri-diagonal matrix from the first-order autoregressive process, and explicit equations exist. Letting $\hat{N}_k = \sum_j r_{jk}$, then $q(V_k)$ is updated by are $a_k = 1 + \hat{N}$, $b_k = \hat{\beta} + \sum_{k'=k+1}^{K} \hat{N}'_k$. For $q(\zeta_{jk})$, the parameters are updated $\eta_{jk} = 1 + \sum_b \text{trace}(\mathbf{R}_{jb}(\hat{\boldsymbol{a}}_{kb}\hat{\boldsymbol{a}}_{kb}^T + \boldsymbol{\Sigma}_{kbb}))$, and $\mu_{jk} = \eta_{jk}^{-1}\sum_b \hat{\boldsymbol{a}}_{kb}^T \mathbf{R}_{jb}\boldsymbol{\nu}_{jb}^{-j}$. The clustering latent variables are updated sequentially by:

$$\log(r_{jk}) \propto -\frac{\tau}{2}\sum_b [(\mu_{jk} + \eta_{jk}^{-1})\text{tr}(\mathbf{R}_{jb}(T_b\boldsymbol{\Sigma}_{kbb} + \hat{\boldsymbol{a}}_{kb}\hat{\boldsymbol{a}}_{kb}^T)) - 2\mu_{jk}(\boldsymbol{x}_b^{-j})^T(\boldsymbol{y}_b \mathbf{R}_{jbb}\hat{\boldsymbol{a}}_{kb})] + \mathbb{E}_q[q(\pi)]$$

$\boldsymbol{x}_b^{-k}$ and $\boldsymbol{y}_b^{-k}$ can be used to calculate $q(\mathbf{A}_k^*)$. The mean of the distribution $q(\mathbf{A}_k)$ is evaluated using the forward filtering-backward smoothing algorithm, and $\boldsymbol{\Sigma}_k^{-1}$ is a block tridiagonal matrix, enabling efficient computations. Further details on updating $q(\mathbf{A}_k^*)$ are found in Section A of the Supplemental Material. Approximating distributions $q(\beta)$, $q(\alpha)$ and $q(\gamma)$ are standard [14, 15].

### 3.1.1 Merge steps

The model is initialized to $K = J$ clusters and the algorithm first finds $q$ for the non-clustering model. This initialization is important because of the superposition measurement model. The algorithm proceeds to merge down to $K'$, where $K'$ is a local mode of the VB algorithm. The procedure is as follows: $(i)$ Randomly choose two clusters $k$ and $k'$ to merge. $(ii)$ Propose a new variational distribution $\tilde{q}$ with $K-1$ clusters. $(iii)$ Calculate the change in the variational lower bound, $\mathcal{L}(\tilde{q}) - \mathcal{L}(q)$, and accept the merge if the variational lower bound increases. As in [12], intelligent sampling of $k$ and $k'$ significantly improves performance. Here, we sample $k$ and $k'$ with weight proportional to $\exp(-\mathcal{K}(\mathbf{A}_k, \mathbf{A}_{k'}; c_0))$, where $\mathcal{K}(\cdot, \cdot; c_0)$ is the radial basis function. In [13] all pairwise clusterings were considered, but that is computationally infeasible in this problem. This approach for merging clusters is similar to that developed in [12].

This algorithm requires *efficient* estimation of the difference in the lower bound. For a proposed $k$ and $k'$, a new variational distribution $\tilde{q}$ is proposed, with $\tilde{q}(z_j = k) = q(z_j = k) + q(z_j = k')$ and $\tilde{q}(z_j = k') = 0$, $\tilde{q}(\beta_k) = \text{Beta}(a_0 + \hat{N}_k + \hat{N}_{k'}, b_0 + \sum_{k^\star=k+1}^{K, k^\star \neq k'} \hat{N}_{k^\star})$, $q(\beta_{k'}) = \delta_0$, and $q(\mathbf{A}_k)$ is calculated. Letting $H(q) = -\sum_j \sum_k r_{jk}\log r_{jk}$, the difference in the lower bound can be calculated:

$$\mathcal{L}(\tilde{q}) - \mathcal{L}(q) = \mathbb{E}_{\tilde{q}}\left[\log p(y|\mathbf{A}_{1,\dots,K}, \boldsymbol{\zeta}, \tau)\frac{p(\mathbf{A}_k|\alpha, \gamma)}{\tilde{q}(\mathbf{A}_k)}\frac{p(\beta_k))}{q(\beta_k)}\right] - H(\tilde{q}) + H(\tilde{p}) \quad (7)$$

$$- \mathbb{E}_q\left[\log p(y|\mathbf{A}_{1,\dots,K}, \boldsymbol{\zeta}, \tau)\frac{p(\mathbf{A}_k|\alpha, \gamma)p(\mathbf{A}'_k|\alpha, \gamma)}{q(\mathbf{A}_k)q(\mathbf{A}'_k)}\frac{p(\beta_k)p(\beta_{k'})}{q(\beta_k)q(\beta_{k'})}\right] + H(q) - H(p)$$

Explicit details on the calculations of these variables are found in Section A of the Supplementary Material, and the block tridiagonal nature of $\boldsymbol{\Lambda}_k$ allows the complete calculation of this value in $\mathcal{O}(BT_b((\hat{N}_k + \hat{N}_{k'}) + L^3))$. This is linear in the amount of data used in the model. The algorithm is stopped after 10 merges in a row are rejected.

### 3.2 Integrated Nested Laplacian Approximation for the Non-Clustering Model

The VB inference method assumes a separable posterior. In the non-clustering model, Integrated Nested Laplacian Approximation (INLA) [16] was used to estimate of the joint posterior, without

| Animal | Invariant | Non-Cluster | Clustering | Animal | Invariant | Non-Cluster | Clustering |
|--------|-----------|-------------|------------|--------|-----------|-------------|------------|
| 1 | 0.1394 | 0.1968 | **0.2094** | 7 | 0.1385 | **0.2567** | 0.2442 |
| 2 | 0.1465 | **0.2382** | 0.2340 | 8 | 0.0902 | **0.3440** | 0.3182 |
| 3 | 0.2251 | 0.3050 | **0.3414** | 9 | 0.1597 | 0.1881 | **0.2362** |
| 4 | 0.0867 | 0.1433 | **0.1434** | 10 | 0.0311 | 0.0803 | **0.0865** |
| 5 | 0.1238 | 0.1867 | **0.1882** | 11 | 0.675 | 0.1064 | **0.1161** |
| 6 | 0.0675 | **0.1407** | 0.1351 | | | | |

Table 1: Mean held-out RFE of the multi-cell models predicting the Hippocampus LFP. "Invariant" denotes the time-invariant model, "Non-cluster" and "clustering" denote the dynamic model without and with clustering.

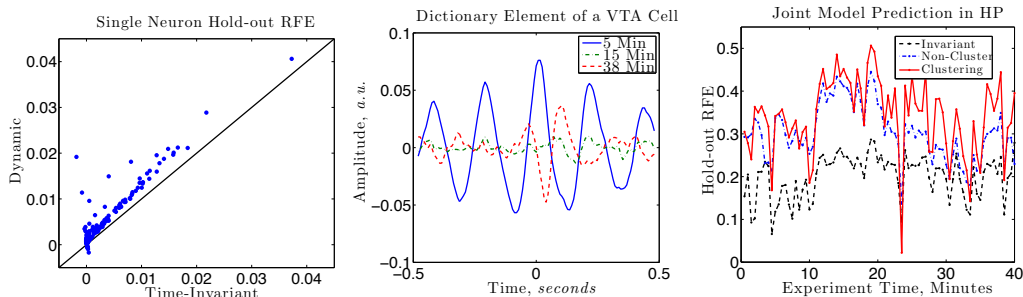

Figure 1: (Left) Mean single-cell holdout RFE predicting mouse 3's Nucleus Accumbens LFP comparing the dynamic and time-invariant model. Each point is a single neuron. (Middle) Convolutional dictionary for a VTA cell predicting mouse 3's Nucleus Accumbens LFP at 5 minutes, 15 minutes, and 38 minutes after the experiment start. (Right) Hold-out RFE over experiment time with the time-invariant, non-clustering, and the clustering model to predict mouse 3's Hippocampus LFP.

assuming separability. Comparisons to INLA constitute an independent validation of VB, for inference in the non-clustering version of the model. The INLA inference procedure is detailed in Supplemental Section B. INLA inference was found to be significantly slower than the VB approximation, so experimental results below are shown for VB. The INLA and VB predictive performance were quantitatively similar for the non-clustering model, providing confidence in the VB results.

## 4 Experiments

### 4.1 Results on Mice Introduced to a Novel Environment

This data set is from a group of 12 mice consisting of male *Clock*-$\Delta$19 (mouse numbers 7-12) and male wild-type littermate controls (mouse numbers 1-6) (further described in [7]). For each animal, 32-48 total microwires were implanted, with 6-16 wires in each of the Nucleus Accumbens, Hippocampus (HP), Prelimbic Cortex (PrL), Thalamus, and the Ventral Tegmental Area (VTA). LFPs were averaged over all electrodes in an area and filtered from 3-50Hz and sampled at 125 Hz. Neuronal activity was recorded using a Multi-Neuron Acquisition Processor (Plexon). 99-192 individual spike trains (single units) were detected per animal. In this dataset animals begin in their home cage, and after 10 minutes are placed in a novel environment for 30 minutes. For analysis, this 40 minute data sequence was binned into 30 second chunks, giving 80 bins. For all experiments we choose $L$ such that the dictionary element covered 0.5 seconds before and after each spike event.

Cross-validation was performed using leave-one-out analysis over time bins, using the error metric of reduction in fractional error (RFE), $1 - ||\boldsymbol{x}_b - \hat{\boldsymbol{x}}_b||_2^2 / ||\boldsymbol{x}_b||_2^2$. Figure 1(left) shows the average hold-out RFE for the time-invariant model and the dynamic model for single spike train predicting mouse 3's Nucleus Accumbens, showing that the dynamic model can give strong improvements on the scale of a single cell (these results are typical). The dynamic model has a higher hold-out RFE on 98.4% of detected cells across all animals and all regions, indicating that the dynamic model generally outperforms the time-invariant model. A dynamic dictionary element from a VTA cell predicting mouse 3's Nucleus Accumbens is shown in Figure 1(middle). At the beginning of the experiment, this cell is linked with a slow, high-amplitude oscillation. After the animal is initially placed into a new environment (illustrated by the 15-minute data point), the amplitude of the dictionary element drops close to zero. Once the animal becomes accustomed to its new environment (illustrated by the 38-minute data point), the cell's original periodic dictionary element begins to appear again. This example shows how cells and LFPs clearly have time-evolving relationships.

The leave-one-out performance of the time-invariant, non-clustering, and clustering models predicting animal 3's Hippocampus LFP with 182 neurons is shown in Figure 1(right). These results show

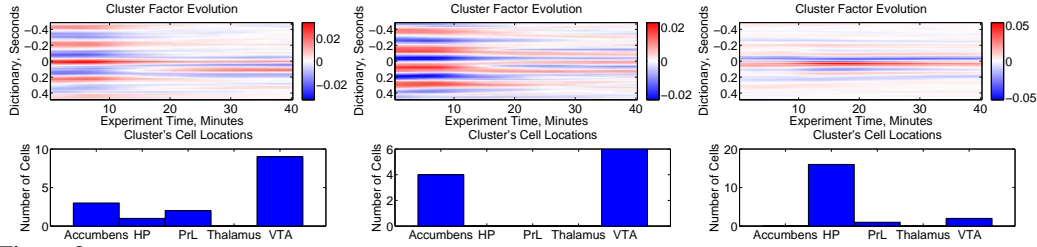

Figure 2: Example clusters predicting mouse 3's Hippocampus LFP. The top part shows the convolutional factor throughout the duration of the experiment, and the bottom part shows the location of the cells in the cluster. Some of the clusters are dynamic whereas others were consistent through the duration of the experiment.

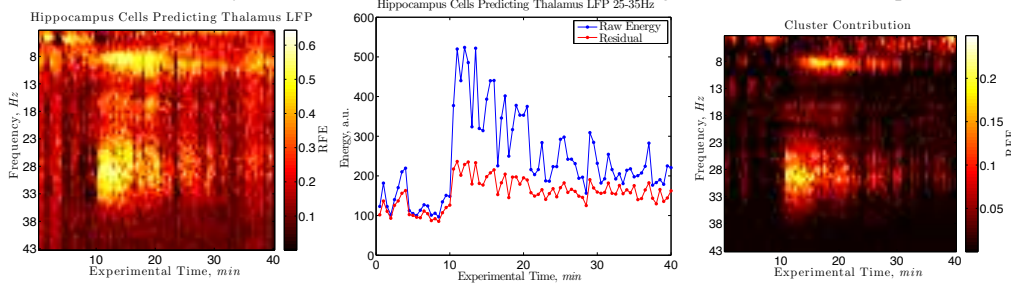

Figure 3: (Left) RFE as a function of time bin and frequency bin for all Hippocampus cells predicting the Thalamus LFP. There is a change in the predictive properties around 10 minutes. (Middle) Total energy versus the unexplained residual for the Hippocampus cells predicting the Thalamus LFP for the frequency band 25-35 Hz. (Right) RFE using only the cluster of cells shown in Figure 2(right).

that predictability changes over time, and indicate that there is a strong increase in LFP predictability when the mouse is placed in the novel environment. Using dynamics improves the results dramatically, and the clustering hold-out results showed further improvements in hold-out performance. The mean hold-out RFE results for the Hippocampus for 11 animals are shown in Table 1 (1 animal was missing this region recording). Results for other regions are shown in Supplemental Tables 1, 2, 3, and 4, and show similar results.

In this dataset, there is little quantitative difference between the clustering and non-clustering models; however, the clustering result is much better for interpretation. One reason for this is that spike-sorting procedures are notoriously imprecise, and often under- or over-cluster. A *clustering* model with equivalent performance is evidence that many neurons have the same shapes and dynamics, and repeated dynamic patterns reduces concerns that dynamics are the result of failure to distinguish distinct neurons. Similarly, clustering of neuron shapes in a single electrode could be the result of over-clustering from the spike-sorting algorithm, but clustering across electrodes gives strong evidence that truly different neurons are clustering together. Additionally, neural action potential shapes drift over time [6, 17], but since cells in a cluster come from different electrodes and regions, this is strong evidence that the dynamics are not due to over-sorting drifting neurons.

Each cluster has both a dynamic shape result as well as well as a neural distribution over regions. Example clustering shapes and histogram cell locations for clusters predicting mouse 3's Thalamus LFP are shown in Figure 2. The top part of this figure shows the base dictionary element evolution over the duration of the experiment. Note that both the (left) and (middle) plots show a dynamic effect around 10 minutes, and the cells primarily come from the Ventral Tegmental Area. The (right) plot shows a fairly stable factor, and its cells are mostly in the Hippocampus region.

The ability to predict the LFP constitutes functional connectivity between a neuron and the neuronal circuit around the electrode for the LFP [18]. Neural circuits have been shown to transfer information through specific frequencies of oscillations, so it is of scientific interest to know the functional connectivity of a group of neurons as a function of frequency [5]. Frequency relationships were explored by filtering the LFP signal after the predicted signal has been removed, using a notch filter at 1 Hz intervals with a 1 Hz bandwidth, and the RFE was calculated for each held-out time bin and frequency bin.

All cells in the Thalamus were used to predict each frequency band in mouse 3's Hippocampus LFP, and this result is shown in Figure 3(left). This figure shows an increase in RFE of the 25-35 Hz band after the animal has been moved to a new location. The RFE on the band from 25-35 Hz is shown

| Region | PrLCx | MOFCCx | NAcShell | NAcCore | Amyg | Hipp | V1 | VTA |
|---|---|---|---|---|---|---|---|---|
| Time-Invariant | 0.1055 | 0.1304 | 0.0904 | 0.1076 | 0.0883 | 0.2091 | 0.1366 | 0.1317 |
| Non-Clustering | 0.1686 | 0.1994 | 0.1599 | 0.1796 | 0.1422 | 0.2662 | 0.1972 | 0.1907 |
| Clustering | 0.1749 | 0.2029 | 0.1609 | 0.1814 | 0.1390 | 0.2798 | 0.2020 | 0.1923 |

| Region | Subnigra | Thal | LHb | DLS | DMS | M1 | OFC | FrA |
|---|---|---|---|---|---|---|---|---|
| Time-Invariant | 0.1309 | 0.1550 | 0.1240 | 0.1237 | 0.1518 | 0.1350 | 0.1878 | 0.1164 |
| Non-Clustering | 0.1939 | 0.2188 | 0.1801 | 0.1973 | 0.2363 | 0.2034 | 0.2695 | 0.1894 |
| Clustering | 0.1950 | 0.2204 | 0.1813 | 0.2012 | 0.2378 | 0.2080 | 0.2723 | 0.1912 |

Table 2: Mean held-out RFE of the animal going through sleep cycles in each region.

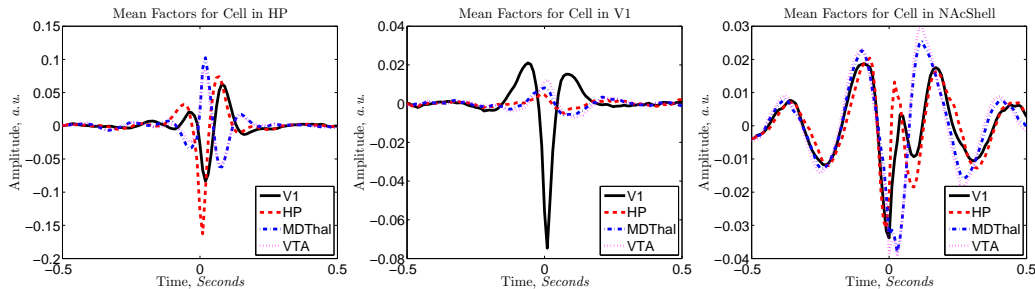

Figure 4: The predictive patterns of individual neurons predicting multiple regions. (Left) A Hippocampus cell is the best single cell predictor of the V1 LFP (Middle) A V1 cell with a relationship only to the V1 LFP. (Right) A Nucleus Accumbens Shell cell that is equivalent in predictive ability to the best V1 cell.

in Figure 3(middle), and shows that while the raw energy in this frequency band is much higher after the move to the novel environment, the cells from the Hippocampus can explain much of the additional energy in this band. In Figure 3(right), we show the same result using only the cluster in Figure 2. Note that there is a change around 10 minutes that is due to both a slight change in the convolutional dictionary and a change in the neural firing patterns.

## 4.2 Results on Sleep Data Set

The second data set was recorded from one mouse going through different sleep cycles over 6 hours. 64 microwires were implanted in 16 different regions of the brain, using the Prelimbix Cortex (PrL), Medial Orbital Frontal Cortex (MOFCCx), the core and shell of the Nucleus Accumbens (NAc), Basal Amygdala (Amy), Hippocampus (Hipp), V1, Ventral Tegmental Area (VTA), Substantia nigra (Subnigra), Medial Dorsal Thalamus (MDThal), Lateral Habenula (LHb), Dorsolateral Striatum (DLS), Dorsomedial Striatum (DMS), Motor Cortex (M1), Orbital Frontal Cortex (OFC), and Frontal Association Cortex (FrA). LFPs were averaged over all electrodes in an area and filtered from 3-50Hz and sampled at 125Hz, and $L$ was set to 0.5 seconds. 163 total neurons (single units) were detected using spike sorting, and the data were split into 360 1-minute time bins. The leave-one-out predictive performance was higher for the dynamic single cell model on 159 out of 163 neurons predicting the Hippocampus LFP. The mean hold-out RFEs for all recorded regions of the brain are shown in Table 2 for all models, and the clustering model is the best performing model in 15 of the 16 regions.

Previously published work looked at the predictability of the V1 LFP signal from individual V1 neurons [11,18,19]. Our experiments find that the dictionary elements for all V1 cells (4 electrodes, 4 cells in this dataset) are time-invariant and match the single-cell time-invariant dictionary shape of [11]. The dictionary elements for a single V1 cell predicting multiple regions are shown in Figure 4(middle; for simplicity, only a subset of brain regions recorded from are shown). This suggests that the V1 cell has a connection to the V1 region, but no other brain region that was recorded from in this model. However, cells in other brain regions showed functional connectivity to V1. The best individual predictor is a cell in the Hippocampus shown in Figure 4(left). An additional example cell is a cell in the Nucleus Accumbens shell that has the same RFE as the best V1 cell, and its shape is shown in Figure 4(right).

Sleep states are typically defined by dynamic changes in functional connectivity across brain regions as measured by EEG (LFPs recorded from the scalp) [20], but little is known about how single neurons contribute to, or interact with, these network changes. To get sleep covariates, each second of data was scored into "awake" or "sleep" states using the methods in [21], and the sleep state was averaged over the time bin. We defined a time bin to be a sleep state if $\geq 95\%$ of the individual sec-

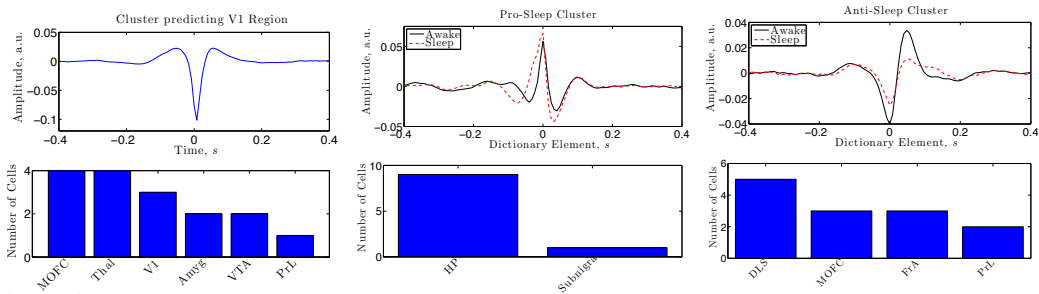

Figure 5: (Left) The cluster predicting the V1 region of the brain, matching known pattern for individual V1 cells [11, 18]. (Middle,Right) Clusters predicting the motor cortex that show positive (pro) and negative (anti) relationships between amplitude and sleep.

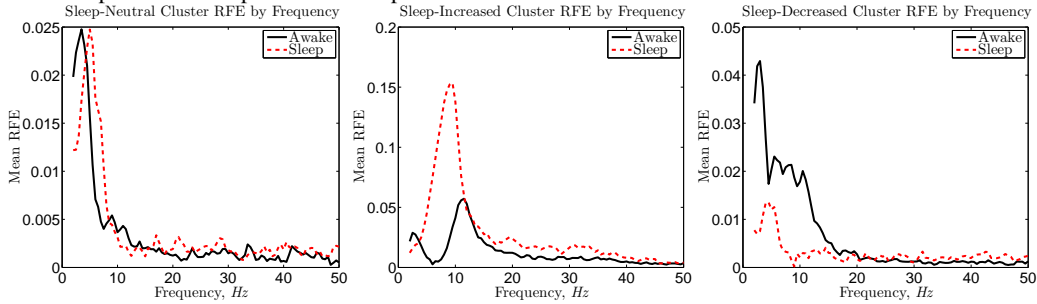

Figure 6: Mean RFE when the animal is awake and when it is asleep. (Left) Cluster's convolution factor is stable, and shows only minor differences between sleep and awake prediction. (Middle and Right) Clusters shown in Figure 5 (left and right), depicting varying patterns with the mouse's sleep state

onds are scored as a sleep state, and the animal is awake if $\leq 5\%$ of the individual seconds are scored as a sleep state. In Figure 5(middle) we show a cluster that is most strongly positively correlated with sleep (pro-sleep), and in Figure 5(right) we show a cluster that is most negatively correlated with sleep (pro-awake). Both figures show the neuron locations as well as the mean waveform shape during sleep and wake. In this case, the pro-sleep cluster is dominantly Hippocampus cells and the anti-sleep cluster comes from many different regions. There may be concern that because these are the maximally correlated clusters, that these results may be atypical. To address this concern, the p-value for finding a cluster this strongly correlated has a p-value $4 \times 10^{-6}$ for Pearson correlation with the Bonferroni correction for multiple tests. Furthermore, 4 of the 25 clusters detected showed correlation above .4 between amplitude and sleep state, so this is not an isolated phenomena.

The RFE changes as both a function of frequency and sleep state for some clusters of neurons. Using 1Hz bandwidth frequency bins, in Figure 6 (middle and right) we show the mean RFE using only the clusters in Figure 5 (middle and right). The cluster associated positively with sleeping shifts its frequency peak and increases its ability to predict when the animal is sleeping. Likewise, the sleep-decreased cluster performs worst at predicting when the animal is asleep. For comparison, in Figure 6 (left) we include the frequency results for cluster with a stable dictionary element. The total RFE is comparable and there is a not a dramatic shift in the peak frequency between the sleep and awake states.

## 5   Conclusions

Novel models and methods are developed here to account for time-varying relationships between neurons and LFPs. Within the context of our experiments, significantly improved predictive performance is realized when one accounts for temporal dynamics in the neuron-LFP interrelationship. Further, the clustering model reveals which neurons have similar relationships to a specific brain region, and the frequencies that are predictable in the LFP change with known dynamics of the animal state. In future work, these ideas can be incorporated with attempts to learn network structure, and LFPs can be considered a common input when exploring networks of neurons [19, 22, 23]. Moreover, future experiments are being designed to place additional electrodes in a single brain region, with the goal of detecting 100 neurons in a single brain region while recording LFPs in up to 20 regions. The methods proposed here will facilitate exploration of both the diversity of neurons and the differences in functional connectivity on an individual neuron scale.

**Acknowlegements**   The research reported here was funded in part by ARO, DARPA, DOE, NGA and ONR. We thank the reviewers for their helpful comments.

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
