[Reviews · NeurIPS 2014]

Submitted by Assigned_Reviewer_1

The author(s) have determined the dynamics of functional connectivity between individual neurons and the local field potential (LFP) across different brain regions. Their work extends previous work to account for the time-varying relationships between action potentials in one area and the LFP in that vicinity or a distant brain area. The work is of high quality and clearly explained. It is not completely original as it extends previous work in this area, but it has the potential to be quite significant.

1) The RFE error metric is interesting and useful but I would also like to see a simpler metric such as a direct measurement of how much of the LFP can be predicted correctly using this model?
2) I do not follow the logic on page 6 line 301-308 — could the authors expand and/or describe the opposite situation and why they think that is not the case?
3) I disagree with the interpretation of Fig 6. page 8 line 413 says that there is not a dramatic shift in 6a but claims there is a shift in 6b. The shift in the peak in 6a appears to be from 4 to 6 hz while the shift in in the peak in 6b appears to be from 9 to 11 hz, which is not different. What is quite different is the change in amplitude. However, the author(s) specifically searched for a sleep-increased cluster, so it is circular logic that the sleep-increased cluster increases during sleep. I imagine the authors are simply trying to point out that sleep-increased clusters do exist, but what is the chance of observing this randomly among however many clusters they searched to find it?

Minor
page 6 line 310 repeated “As well”
page 6 line 317 sentence ending with ref 5. This sentence is still speculative, and I would suggest a better reference that more directly shows that INFORMATION is transferred via oscillations.
Summary: The authors have extended previous work looking at the functional connectivity between single units and the LFP signal to the time domain, uncovering interesting dynamics in the relationship across different brain regions when animals are introduced to a novel environment and between sleep/awake states.

Submitted by Assigned_Reviewer_23

Summary:

The authors propose a convolutional dictionary model that captures relationships between spikes and LFP signals across multiple brain regions. Significant contributions are 1) the use of a clustering prior on the dictionary elements, 2) the ability to model dynamics of the dictionary elements, and 3) the corresponding mean-field variational inference procedure for the model.

The authors then present a multitude of results on two large datasets. They find that time-varying dictionary elements yield superior performance over static elements, and that the use of clustering allows for improved interpretability of the data.

In the novel environment experiment, they find that some dictionary elements change at the introduction to the new environment while others do not – and that the clustering reveals, for example, that a particular stable element is mostly associated with HP cells.

Quality:

The authors show convincingly that clustering and dictionary dynamics offer significant improvements both in quantitative fit and qualitative interpretability over simpler spiking-to-LFP models. A strong point of the paper is the variety of results presented on both datasets, showcasing the utility of the method.

The model choices are reasonable. For example, the observation that similar results were found for a wide range of Bs suggest that the choice of modeling the dynamics over large discrete bins is sensible.

With that said, what motivation is there for a first-order AR process over higher orders? This seems a stringent limitation on the dynamics of the dictionary elements.

The paper might benefit from a discussion on how both spiking and LFP signals change over the course of both experiments. That is, are the dynamics of the dictionary elements due to changes in spiking statistics, LFP signals, or both? The emphasis was on the changes in dictionary elements themselves, illustrating that it is the relationship between spikes and LFP that changes over the course of the experiment, but it might help to understand this point in relation to the spiking and LFP signals themselves.

Clarity:

This paper is exceptionally clear.

Small details:
- There is a mismatch in the text on page 6 and the corresponding figures 2 and 3 (text says thalamus LFP, figure says HP LFP, etc)
- best values not bolded in table 2.
- how many clusters are typically identified. E.g. figure 2 shows 3 of how many?

Originality:

The method is original, and the authors present several original types of analyses one could perform with the method (e.g. see results in figure 3).

Significance:

This work appears to offer significant improvements over previous methods, and has potential to see widespread use as multi-region datasets become more common.
Summary: A strong and thorough contribution that has potential to see widespread use in LFP analysis.

Submitted by Assigned_Reviewer_39

The authors developed a novel approach to account for time-varying relationship between LFP and spike trains across multiple brain areas simultaneously recorded using multi-electrode arrays in mouse. They found that by allowing the temporal kernel to vary over time, they can improve the predictability of the LFP based on the spiking activities of neurons in the same area or from different areas. Insofar as predictability can be considered as a measure of functional connectivity, these findings suggest the connectivity of neurons and neural circuits across different brain areas vary dynamically in interesting ways. Although the meaning and significance of these findings are not clear at this stage and require further investigation, the approach, including the idea of using dynamic temporal kernel and clustering neurons together to predict LFP, is novel and interesting and would be of interest to the NIPS community.
Summary: A novel approach for characterizing the dynamic relationship between different brain areas by characterizing the relationship between LFP and spike trains in different areas, and by clustering similar neurons (by forcing them to have similar temporal kernel) to maximize model prediction. Although the interpretation of the effects discovered by this method is not clear at this point, the approach would be of interest to the community.
Author Feedback
Author rebuttal: We thank the reviewers for their helpful comments. The minor comments will be addressed in the revised paper, and specific larger concerns are discussed below (and will also be addressed in the revised paper).

Response to reviewer 1's comments:
To address concerns on the full prediction of the signal, this model reduces the hold-out MSE of the full filtered signal by up to 35% in the experiments, but this value is dependent on the filtering procedure. Since different filtering bandwidths gives different results, the RFE was shown as function of the specific filtering that was used and as a function of frequency bins and time.

In lines 301-308, the concern addressed is the uncertain nature of spike sorting. In the under-sorting case, such as when real neurons A and B get sorted into a single detected neuron C, it's unclear whether dynamics on C are due to (i) actual dynamics of a single neuron, or (ii) differing evolution of firing rates between neurons A and B causing the combined neuron C to appear to have dynamics. While under-sorting is occasionally inevitable in spike sorting, similar patterns in sorted neurons across multiple electrodes provides stronger evidence that the dynamics are real. Similarly, clustering of neuron shapes in a single electrode could be the result of over-clustering from the spike-sorting algorithm, but clustering across electrodes gives strong evidence that truly different neurons are clustering together. This will be clarified in the revised paper.

Addressing Figs. 6a and 6b, we meant to convey the difference in amplitude more than the difference in peak frequency, and the paragraph will be adjusted to be clearer on this point. We feel it is unlikely to find clusters whose amplitude correlate this strongly with sleep states by chance. In this experiment, the Pearson correlation between the sleep state and the cluster’s amplitude gives a two-tailed p-value of 1.6*10^-7. A Bonferroni correction for the search over 25 clusters would give an adjusted p-value of 4*10^-6. Furthermore, 4 of the 25 clusters detected showed correlation above .4 between amplitude and sleep state, so this is not an isolated phenomena.

Response to reviewer 23's comments:
Since these systems are quite noisy, the first-order AR process gives simple-to-interpret results that are less prone to overfitting. Future work will include exploring more flexible dynamics, and the clustering model allow neurons to share strength to better learn the dynamics.

The relationship between dynamics in the neural firing rates and LFP signals and the shape of the dictionary elements is unclear in these datasets. In the novel environment experiment, neurons showed no upward or downward trend in total firings per time bin, and the standard deviation on total fires in each bin was typical less than 20% of the mean firing rate per neuron. The dynamics here are not due to drastically changing firing rates. Dynamics in the LFP, however, appear to have a slight effect on dictionary element shape, as can be seen in Figure 3(middle), but are far from completely explaining the dynamics. Furthermore, some dictionary element shape dynamics don’t correspond to changes in LFP spectral power.

There are between 10-30 clusters depending on the experiment.